# Matrix Metalloproteinases in Cardioembolic Stroke: From Background to Complications

**DOI:** 10.3390/ijms24043628

**Published:** 2023-02-11

**Authors:** Anna Wysocka, Jacek Szczygielski, Marta Kopańska, Joachim M. Oertel, Andrzej Głowniak

**Affiliations:** 1Department of Internal Medicine and Internal Nursing, Faculty of Health Sciences, Medical University of Lublin, 20-093 Lublin, Poland; 2Department of Neurosurgery, Institute of Medical Sciences, University of Rzeszów, 35-310 Rzeszów, Poland; 3Department of Neurosurgery, Saarland University Medical Center, Saarland University Faculty of Medicine, 66421 Homburg, Saar, Germany; 4Department of Physiology, Institute of Medical Sciences, University of Rzeszów, 35-310 Rzeszów, Poland; 5Department of Cardiology, Medical University of Lublin, 20-954 Lublin, Poland

**Keywords:** matrix metalloproteinase, stroke, thrombembolism, atrial fibrillation

## Abstract

Matrix metalloproteinases (MMPs) are endopeptidases participating in physiological processes of the brain, maintaining the blood–brain barrier integrity and playing a critical role in cerebral ischemia. In the acute phase of stroke activity, the expression of MMPs increase and is associated with adverse effects, but in the post-stroke phase, MMPs contribute to the process of healing by remodeling tissue lesions. The imbalance between MMPs and their inhibitors results in excessive fibrosis associated with the enhanced risk of atrial fibrillation (AF), which is the main cause of cardioembolic strokes. MMPs activity disturbances were observed in the development of hypertension, diabetes, heart failure and vascular disease enclosed in CHA_2_DS_2_VASc score, the scale commonly used to evaluate the risk of thromboembolic complications risk in AF patients. MMPs involved in hemorrhagic complications of stroke and activated by reperfusion therapy may also worsen the stroke outcome. In the present review, we briefly summarize the role of MMPs in the ischemic stroke with particular consideration of the cardioembolic stroke and its complications. Moreover, we discuss the genetic background, regulation pathways, clinical risk factors and impact of MMPs on the clinical outcome.

## 1. Introduction

Matrix metalloproteinases (MMP) are a family of proteolytic enzymes, which are characterized by the conservative structure, and exhibit various functions both in physiological as well as in pathological conditions. MMPs are capable of cleaving the peptide bindings in the majority of extracellular matrix compounds such as fibronectin, laminin, proteoglicans, various types of collagen and tight junction proteins. They play an important role in the remodeling of connective tissue and regulate cell-to-cell interactions. [1,2,3,4]. MMPs are produced inside and secreted outside the cells of which the connective tissue is built, including fibroblasts, osteoblasts, vascular smooth muscle and endothelial cells. Additionally, macrophages, neutrophils and lymphocytes involved in inflammatory process, as well as placental cells such as cytotrophoblasts, synthesize and secrete MMPs [4,5]. The majority of MMPs predominantly act extracellularly, but some were detected also in nuclear extracts from cardiac myocytes and hepatocytes, where they play a role in DNA-repairing process [3,4]. The appropriate function of MMPs is crucial in such physiological conditions as connective tissue turnover, proliferation and differentiation of cells, embryogenesis, angiogenesis and apoptosis. Hence, MMPs overexpression or deregulation were observed in multiple pathological conditions including metastatic process, excessive fibrosis or tissue destruction [4,5,6].

Ischemic stroke (IS), contributing to 80–90% of all stokes, results from limited cerebral blood flow caused by occlusion or narrowing blood vessels. IS accounts for several millions of deaths per year and is ranked as being among the two main causes of death worldwide. Approximately one third of all IS cases are cardioembolic strokes (CES), resulting from the cerebral artery occlusion by the thrombus of cardiac origin. The source of thromboembolism in patients suffering from atrial fibrillation (AF) is mainly left atrial appendage [7,8].The risk of stroke in patients with AF is increased five-fold compared to the general population. Moreover, AF-related strokes are associated with worse outcomes—increased mortality and disability [7,9]. In AF patients, the risk of stroke may be reduced by oral anticoagulation, but the balance between the risk of stroke and the potential risk of major bleeding related with anticoagulation should be carefully evaluated. The risk of stroke in AF patients is estimated with the commonly implemented CHA_2_DS_2_VASc score including age, sex, history of cardiac heart failure, arterial hypertension, diabetes mellitus, stroke/TIA/thromboembolism or vascular disease (prior myocardial infarction, peripheral artery disease or aortic plaque). Most of these factors, such as elderly age, hypertension, diabetes and valvular heart disease, are also recognized as risk factors of AF occurrence [10]. On the other hand, even relatively young patients, without comorbidities, with a clinically “low risk” of stroke, thromboembolic events affecting the central nervous system defined as an embolic stroke of undetermined source (ESUS) still occur [11]. Hence, the identification of new additional markers, laying in the background of pathophysiology of both AF and cerebral arteries thromboembolism is important, and MMPs can be considered amongst these markers.

The aim of our review is to present briefly the role of MMPs in the cardioembolic stroke, focusing on clinical features and taking into consideration the wide context from the genetic and molecular background of stroke risk factors up to stroke complications. We performed a literature search with the use of the PubMed database to identify relevant articles. Initially, the terms “matrix metalloproteinase” and “cardioembolic stroke” were used with limiting the data availability to “full text” and publication date to 10 years, which allowed to identify only 32 full text articles. After broadening the searching to “MMP” and “ischemic stroke”, 496 results were obtained, amongst from which texts appropriate for preparing this review were chosen. An additional search was performed using the following terms: “matrix metalloproteinase” and, respectively, “atrial fibrillation”, “myocardial infarct”, “arterial hypertension”, “heart failure”, “diabetes” and “matrix metalloproteinase polymorphism” as well as all of the above-mentioned terms. Additionally, appropriate papers from the reference lists of previously selected articles were included. Only papers written in English were considered.

The reviewed information has been arranged in two following sections: (1) structure and regulation of MMPs, (2) role of matrix MMPs in stroke, subdivided into four subsections presenting the association of MMPs with (i) AF (ii) risk factors of stroke in AF, (iii) stroke occurrence and healing and iv) role of MMPs in stroke complications. Finally, the considerations conducted in this review have been summarized into three brief conclusions.

## 2. Structure and Regulation of MMPs

The first description of enzymes, which facilitate the tumor’s growth, was performed over 70 years ago and since that time, twenty-eight MMPs were purified and characterized in vertebrates, of which twenty-three were isolated in humans [4,5]. All MMPs in vertebrates were assigned subsequent numbers, but MMP-4, MMP-5, MMP-6 and MMP-22 were identified as being identical to other members [6]. MMPs, together with desintegrines and metalloproteinases (ADAMs) and ADAMs with thrombospondine motif (ADAMTS), belong to the metzincs superfamily [2,4,5,6]. All of them are multidomain zinc-dependent endopeptidases with common, highly conserved, zinc binding motif HExxHxxGxxH. Typically, in MMPs, this domain is HEBGHxLGLxHSBMxP [4,5,12]. The basic structure of all MMPs is very similar and most of them consist of five elements. In the N-terminus of the peptide chain, the signal peptide (predomain) is localized, which is characterized by variable length and enables the enzyme to transport into the endoplasmic reticulum and enables secretion outside the cell [4,12]. The prodomain, which consists of about 80 amino acids, includes the sulfur bridge and is connected with the predomain, and it keeps MMPs inactive until it is cleaved [1,4,12,13]. A spherical catalytic domain with a diameter of 40A consists of about 160 amino acids, which are shaped into five sheets and three helices connected by eight loops. The particle contains three calcium ions. Apart from catalytic zinc ion localized in the active site in the center of a cavity on the surface of protein, there is another Zn^2+^ in the catalytic domain, which has structural function. Furthermore, three calcium ions, crucial for maintaining protein structure, play the same role [4,14,15,16,17]. Three histidine residues in the common structural motif VAAHEXGHXXGXXH interrelate with the catalytic zinc ions. The cysteine residue in PRCGXPD motif within the propeptide has the same function [1,13]. The most variable motif within the catalytic domain is the Ω-loop between the second and third helices, probably responsible for the specificity of different MMPs for various substrates [18]. The second region of the active site is a specific site of S1, a pocket, the outer wall of which is formed by the highly conservative region called “met-turn” (XBMX), located in the terminal part of catalytic domains, common for MMPs, ADAMS and ADAMTS [4,6]. In the carboxyl-terminus of the particle, the hemopectine-like domain (with roughly 200 amino acids and four propellers) is localized, which is responsible for binding into the catalytic center [6,16,17]. The last two structures are connected by a hinge region—a proline-rich linker peptide, which variably for different MMPs, consists of 14–69aminoacids [4,12,16]. The MMP-7 and MMP-26, as well as the MMP-23, are devoid of the linker and hemopexine-like domain; however, MMP-23 possesses additional cystein- and immunoglobulin-like domains, which are both connected with catalytic domain. The individual feature of MMP-23 is also the presence of a signal anchor in the signal peptide position [6,16,17]. In the catalytic domain of MMP-2 and MMP-9,fibronectine type II motif, repeated three times, is present [4]. In MMP-14, -15, -16 and -24, unlike in the others, a transmembrane domain together with small C-terminal domain localized in cytoplasmatic domain was found [6,16,17].

Regarding the ability of particular compounds of extracellular matrix proteolysis and according to the domains structure, MMPs have been divided into matrylysins, collagenases, gelatinases, stromelysins, membrane type MMPs and other MMPs [4,19]. Matrilysins (MMP-7 and MMP-26) are the smallest particles in MMPs family and do not contain the hemopexin-like domain. Collagenases (MMP-1, MMP-8 and MMP-13) and stromelysins (MMP-3, MMP-10 and MMP-11) are built from the catalytic domain and hemopexine-like domain. A complex collagen-binding section called fibronectin-like domain is localized within the catalytic domain of gelatinases (MMP-2 and MMP-9). Some MMPs [MMP-11, MMP-23, MMP-28 and membrane type MMPs (MT-MMPs)] contain an additional sequence, which is recognized by furin, converting the enzyme and allowing secretion in the active form. MT-MMPs MMP-14, MMP-15, MMP-16, MMP-24 and MMP-25 are anchored in the cell membrane by atransmembranic domain and possess cytoplasmatic tail. According to a bioinformatic analysis, five types of MMPs including (1) non-furin regulated MMPs (1,3,7,8,12,13,20 and 27), (2) MMPs containing three fibronectine-like elements in the catalytic domain (2 and 9), (3) glycophosphatydyloinositol-anchored MMPs (17 and 25), (4) MMPs containing a transmembrane domain (14,15 and 24) and (5) other MMPs (19,21,23 and 28) were distinguished [20].

The activity of MMPs is regulated at several levels: from the gene expression, through secretion as inactive forms of zymogenes and stepwise activation, up to specific endogenous inhibitors action [21]. The majority of MMPs genes are not expressed permanently but are transcribed after the cell activation by cytokines, growth factors, hormones as well as cell–cell and cell–matrix interactions [4,5,6,15,17]. For example, *MMPs* expression is physiologically activated by estrogen and progesterone [5,15]. In an animal model, it was shown that *MMP-2*expression is activated by platelet-derived growth factor-BB (PDGF-BB) through Rho-associated protein kinase (ROCK) and ERK/p38 MAPK (extracellular signal-regulated kinases/p38 mitogen activated protein kinase) pathways [22]. Another growth factor upregulating *MMP-1* and *MMP-9* expression in carotid arteries atherosclerotic lesions is epidermal growth factor (EGF), which also was reported as an enhancer of MMP-9 activity in vascular smooth muscle cells [23]. On the contrary, transforming growth factor-β1 (TGF-β1), through inhibitory element present in the promoter region of majority of MMPs genes, suppresses these genes expression, except from the MMP-2 gene, which is resistant to downregulation by TGF-β1 [5,24].

Another important regulation level is MMPs production and secretion in the form of inactive pre-proenzymes. After the translation process, the signal peptide is cleaved and the propeptide (enzymatically inactive zymogen) is secreted into extracellular medium. In the next step, after the disruption of the bound between the cysteine localized in prodomain and zinc ion in the catalytic center, the active enzyme is generated [4,5,6]. Tissue and plasma serine proteinases are endopeptidases, capable of removing the part of prodomain in the proteinase-susceptible “bait” region, which is the first stage of the intermolecular processing and stepwise enzyme activation. Complete elimination of the propeptide is achieved due to activity of other MMPs or MMP intermediates [1,4,6,12,13,14,15]. Several pro-MMPs at the end of the propeptide include the sequence RX[K/R]R recognized by furin-like convertase, activated inside the cells and secreted in the form of active enzymes [6,25,26]. The regulation of these MMPs occurs as a result of tissue-specific localization, the action of proteolysis inhibitors or, asin the case of MT-1 MMP, rapid endocytosis and only limited recycling outside the cell membrane [6,25]. In the pro-MMP-2 activation process on the cell surface, two particles of MT-1 MMP (MMP-14) are involved. The first active MT-1 MMP particle binds with the complex of pro-MMP-2 and TIMP-2, which are connected by MMP-2 hemopexin domain and the non-inhibitory C-terminal domain of the TIMP-2, the second allows to form a suitable three-dimensional structure and serves as an activator of pro-MMP-2 [6,27]. In the laboratory, a unique property of MMPs to be activated by mercury compounds (such as 4-aminophenylmercurin acetate or mercury chloride) and other thiol reagents such as N-ethylmaleimide, chaotropic agents, heat or low pH is used. The mechanism of pro-MMPs activation by the above-mentioned reagents is the disturbance of interaction between the zinc ion and cysteine [4,6]. Pro-MMPs can be likewise activated during inflammatory process with the involvement of oxidants such as peroxinitrite (ONOO-) and hypochlorous acid (HOCl) [6].

Finally, MMP activity is regulated by endogenous inhibitors, which may be divided into two major types: nonspecific and specific tissue inhibitors of metalloproteinases (TIMPs). Non-specific MMPs inhibitors include the α1-proteinase inhibitor, which binds covalently to the most of proteinases, and α2-macroglobulin, which complexes with several proteinases are rapidly removed by receptor- dependent endocytosis [4,5,6,28]. The Reversion Inducing Cysteine Rich Protein with Kazal Motifs (RECK) plays an important role in the regulation of various MMPs, ADAMs and ADAMTS activity. RECK is a protein containing 971 amino acids (of which about 9% are cysteine residues) and anchored to the cell membrane by a glycosylphosphatidylinositol (GPI). The GPI anchor is a fragment of RECK responsible for the ability of regulation membrane-bound proteins such as MT1-MMP, ADAM10, and ADAM17. One of the RECK particle domains, containing amino acids from 635 to 654 with three disulfide bridges between six Cys residues, completely share the structure with the Kazal motifs, belonging to the serine protease inhibitors, whereas two other domains (716–735 and 754–772) partially overlap Kazal motifs. Additionally, it was demonstrated that glycosylation of one out of five asparagine residues localized at the NH2 terminal (Asn297) results in the inhibition of MMP-9 secretion due to pro-MMP9 sequestration, while glycosylation of Asn352 residue leads to prevention of MMP-2 activation. As was demonstrated in cultured cells, RECK acts also at a genetic level and suppresses *MMP-9* transcription. The mechanism of this process is the binding inhibition of two subunits of Activator Protein (AP)-1 to TRE (12-O-tetradecanoylphorbol-13-acetate-responsive element) in the *MMP-9* promoter region. Apart from suppression of MMP-9 and activation of MMP-2, the role of RECK in the inhibition of MMP-7, -14 and -17 was demonstrated [29,30,31,32,33].

In contrast to RECK, despite a partially similar structure and functions, TIMPs (-1,-2, -3, and -4), consisting of 184 to 194 amino acids, are able to control specifically pro-enzymes as well as active MMPs. An N-terminal domain of TIMPs, which similarly to a C-terminal domain contains three disulfide bridges, but forms an independent unit, is necessary to the specific stereochemical binding of TIMP to the MMP molecule. The crucial structure for the MMPs inhibition is the conserved netrin module providing the Zn^2+^ cofactor chelation. The TIMPs 1,2 and 4 are soluble, while TIMP-3 is (similarly to RECK) not soluble, but in contrast to cell-surface anchored RECK, bound to ECM [33]. The inhibition effectiveness of particular TIMPs against various MMPs is different;for example, TIMP-1 poorly inhibits MT1-MMP, MT3-MMP, MT5-MMP and MMP-19, but is highly effective against ADAM-10 [6]. The imbalance between the activity of MMPs and their inhibitors leads to numerous disorders resulting from excessive fibrosis when the activity of TIMPs prevails, or extracellular matrix weakening and tissue degradation in the case of MMPs predomination [4]. The MMPs structure and regulation discussed above are summarized in Figure 1.

## 3. Role of MMPs in Cardioembolic Stroke

### 3.1. Association of MMPs with Atrial Fibrillation

AF development and progression is highly dependent on the, initially electrical and then structural, remodeling of the left atrium (LA). The prolonged rapid pacing of LA leads to the shortening of the atrial refractory period by the inhibition of the transient outward current and the L-type Ca^2+^ current, as demonstrated in experimental studies. The shorter wavelength enables conduction of many coexisting, reentering wavelets within the atrium and this way contributes to arrhythmia persistence [34,35]. The clinical implication of these findings is the fact that atria affected with the AF episode are more susceptible to consecutives arrhythmias, this was summarized with the concept that “AF begets AF”. Further stages of atrial remodeling include biomarkers activation, contractile dysfunction, enlargement of atrium and finally atrial walls fibrosis as a result of extracellular matrix activation, fibroblasts proliferation, oxidative stress, inflammation, apoptosis and necrosis [34,36,37,38]. During atrial fibrosis process, cardiomyocytes, lost as a result of apoptosis and necrosis in conditions of enhanced oxidative stress or inflammation, are replaced by compounds of ECM. More intensive heterogeneity of atrial tissue in conjunction with electrical remodeling provides altered electrical wave propagation, facilitating AF initiation and persistence. In abnormal conditions of blood stasis in the cavity of fibrillating atrium, blood coagulability enhances and consequently increases the risk of thrombus formation. The higher the burden of arrhythmia, the more probable risk of systemic thromboembolism and stroke [37,39,40].

The role of MMPs in the structural remodeling of the LA was evaluated in an experimental study, assessing the response of the canine left atrium to rapid pacing (400 beats/min for 6 weeks). Investigators demonstrated significantly and selectively increased activity of MMP-9, whereas the level of the cardio-specific tissue inhibitor of MMP-4 (TIMP-4) and MMP complex in LA myocardium in dogs with pacing-induced atrial failure was significantly decreased in comparison with control animals [41]. It was also demonstrated that human atrial cardiomyocytes are able to express the *MMP-9* gene and secrete MMP-9 protein, and this process is enhanced by rapid pacing in a time- and dose-dependent manner. After transfection of human myocytes with artificially created constructs containing rs3918242 C/T single nucleotide polymorphism (SNP) in *MMP-9* promoter, it was demonstrated that the promoter activity in cells containing rs3918242T variant was increased and additionally enhanced by rapid pacing in comparison with wild-type (C) constructs. Nevertheless, the expected higher susceptibility for AF in Taiwanese patients carrying *MMP-9* rs3918242 T allele was not confirmed. Similarly, no significant differences regarding MMP-9 expression in the atrial tissue of patients with AF carrying the *MMP-9* rs3918242 CT variant was found in comparison with the wild-type CC carriers [42]. In Chinese patients with hypertensive heart disease with or without AF, significant differences in genotype and alleles frequency of -418G/C polymorphism in the *TIMP-2* gene were observed. The occurrence of C allele (genotypes GC and CC) in both groups was associated with decreased levels of TIMP-2 and with the higher risk of AF in comparison with GG genotype independently from age, left atrium size, left ventricle (LV) mass index or antihypertensive treatment. The frequency of MMP-2 polymorphic variants, which was simultaneously investigated, did not differ significantly in patients with and without AF [43]. Contrarily, the distribution of -1562C/T variants of the *MMP-9* gene in this population was different with the higher frequency of T allele (TT and CT genotypes) than CC genotype in patients with AF independently from age, LA size and the treatment with drugs influencing renin-angiotensin system (RAS) [44]. The modification of MMPs’ genes expression, as a result of the polymorphic variants presence, contributes to atrial remodeling and the rate of AF recurrence. In a prospective study evaluating the association of *MMP-1* and *MMP-3* polymorphisms in patients with persistent AF, who had restored sinus rhythm due to electrical cardioversion (ECV), authors observed an increased risk of AF recurrence in carriers of both 5A and 1G alleles in comparison with patients without these alleles [45]. However, in patients with non-valvular AF and a low CHA_2_DS_2_VASc score, who did not receive oral anticoagulants which are reducing the risk of CES, the -1562C/T SNP of *MMP-9* gene (rs3918242) was not identified as the risk factor of CES occurrence [46]. Additionally, data regarding significant differences in extracellular matrix synthesis and degradation markers, obtained in several investigations of patients with AF in various clinical circumstances, were univocal. In atrial cardiomiocytes from patients with paroxsysmal AF, micro RNA-146b-5p (miR-146b-5p), one of miRs post-transcriptionally regulating genes by binding to mRNA, was identified as inhibiting the expression of TIMP-4 and promoting the atrial fibrosis [47]. Patients with AF, independently from the arrhythmia duration (paroxysmal or persistent AF), were characterized by a higher level of TIMP-1 in comparison with individuals with the sinus rhythm (SR), Moreover, patients with persistent AF had lower levels of MMP-1 but increased levels of TIMP-1 in comparison with patients with paroxysmal AF [48]. In comparison with patients with SR, the higher concentration of MMP-9 was found both in patients with persistent AF undergoing ECV (with the significant increase after ECV procedure), as in patients with continuous AF. No difference in the TIMP-1 level between investigated groups was observed [49]. Additionally, MMP-9/TIMP-1 ratio in patients suffering from AF compared with patients with SR were significantly higher and additionally correlated with the levels of NT-pro BNP, IL-6 and LA diameter [50]. Contrarily, in the group of patients with postoperative atrial fibrillation (POAF) after primary coronary artery by-pass grafting, the level of MMP-9 was decreased and TIMP-1 level increased [51]. In the bioptates of right atrium appendages (RAA) and right atrium free walls (RWF) achieved intraoperatively during mini-Maze procedure in patients with persistent AF and patients with SR, in all samples collected from patients with AF, MMP-2, MMP-9 and TIMP-2 levels were significantly elevated in comparison with patients with SR. TIMP-3 and TIMP-4 levels did not differ between the groups, whereas the level of RECK was increased in RFW samples in patients with AF in comparison to RAA samples from patients with AF and with SR [52]. Additionally, the worse effect of pulmonary veins isolation and the higher frequency of AF recurrence after radiofrequency catheter ablation in patients with elevated pre-ablation level of TIMP-2 may be considered as indirect evidence of the important role of collagen turnover markers in atrial remodeling [53]. Authors suggest that changes in the activity of ECM turnover markers are not only associated with AF initiation, but the severity of their deregulation may contribute gradually to the AF burden and progression from paroxysmal to a persistent form of arrhythmia. In the meta-analysis of 33 studies evaluating the association between MMPs and TIMPs levels in blood and/or atrial tissue and AF, it was demonstrated that in patients with AF, the mRNA levels of MMP-1 in samples from atrial tissue were significantly increased and the levels of circulating TIMP-2 decreased in comparison with control group. Additionally, MMP-2 and MMP-9 levels in blood and atrial tissue remain positively associated with AF occurrence, but authors considered these relations to be a result of publication bias [54]. In a prospective case–cohort study evaluating the relationships between collagen turnover markers and AF incidence, the investigators demonstrated that only elevated activity of MMP-9 was independently associated with increased risk of AF [55].On the basis of various studies, a new model has been proposed, assuming that atrial fibrosis or more broadly defined atrial disease resulting from AF is a common thread connecting the arrhythmia and thromboembolic events. This idea may be confirmed by the evidence that deregulation of ECM turnover and enhanced atrial fibrosis were found both in patients with AF with stroke [37].

### 3.2. MMPs in Risk Factors of Stroke in the Course of Atrial Fibrillation

In conditions such as hypertension, diabetes, heart failure or vascular disease (myocardial infarction, previous stroke or atherosclerotic plaques occurrence), which are all stroke risk factors in patients with AF, significant changes in MMPs activity and genetics were found. The increased MMPs activity may cause the degradation of collagen contained in the fibrous cap, which leads to unfavorable atherosclerotic plaque remodeling and destabilization, and may be responsible for the plaque rupture with subsequent thrombus formation—the direct cause of myocardial infarction and stroke [56,57]. In animal studies, the role of MMP-7, -9 -12 as well as TIMP-1 and -2 in the progression and disruption of atherosclerotic plaques in brachiocephalic arteries of apolipoprotein E-deficient (*ApoE*−/−) mice was demonstrated [57,58]. In double knockout animals *ApoE*−/− mice with concomitant deletions in *Mmp-7*; *-9* or *-12* and *Timp-1*), high fat feeding resulted in a higher frequency of sudden death only in *Mmp-7* knockout animals in comparison with wild-type controls. The atherosclerosis burden was decreased in *Mmp-7* and *Mmp-12* knockout animals but enhanced in *Timp-1*-devoid animals, as well as the degree of coronary arteries stenosis, whereas post-infarction cardiac fibrosis was increased in *Mmp-7* as and markedly reduced in *Timp-1*-deficient mice. Authors underlined different effects of MMPs and TIMPs on coronary arteries plaques vulnerability and myocardial fibrosis [59]. In human and animal models, the altered levels of several MMPs (including MMP-1, -2, -3, -7, -8, -9, -12, -13, -14) and all four TIMPs after coronary artery permanent occlusion were observed. In cardiovascular studies, MMP-2 and MMP-9 were the most intensively investigated. In early stages of MI, increased activities of MMP-2 and MMP-9 were reported, suggesting their predominating role in early post-MI response. In the animal models of MI, the MMP-9 mRNA level was elevated 6 h after the vessel occlusion with the peak after 24 h, whereas increased MMP-9 activity was detected from 24 h to 3 days after MI with peaks at 4–7 days and declined after 14 days. Similar profiles of the activity were presented for MMP-1 in rodents, MMP-2 in humans, MMP-3 in both humans and animal models, whereas for MMP-7, MMP-13 and MMP-14, increased activities were reported in the later stages of MI. MMP-8 mRNA increased as early as in 6 h, but MMP-8 protein expression was elevated in 2 weeks and was still detected 16 weeks post-MI [60]. In the early period of MI, MMPs are enrolled predominantly in the process of the myocardial proteins proteolysis and degradation, subsequently MMPs can modulate inflammation and angiogenesis by regulation of ECM and non-ECM substrates (including aggrecan, hyaluronan receptor CD44, complement C1q, connexin 43, fibrinogen, fibronectin, fibroblast growth factor receptor 1 and interleukin-1β), whereas in the later post-MI stages may contribute to excessive fibrosis and adverse remodeling [56,60,61]. The interesting observation was reported in Mmp-7 null mice, in which the levels of connexin-43 were increased following MI. As a result, improved survival rates and favorable electrical parameters of myocardium were observed, which indicate the important role of MMP-7 in post-MI arrhythmia induction [62]. In patients with acute MI, complicated with cerebral ischemia (CI) confirmed in cerebral computed tomography (CT), the levels of MMP-2 and MMP-9 as well as MMP-2 and MMP-9 mRNA expression were significantly higher in comparison with patients with MI, but without CI. In group of patients with MI and CI, the positive correlations between the expression levels of MMP-2 and-9 and the size of CI, as well as the neurological deficit score, were observed [63]. Additionally, SNPs of *MMPs* genes influencing the protein expression and activity were reported as being associated with MI. The -1575 A/G and -1306 T/C SNPs in *MMP-2* gene were described as biomarkers of MI and triple vessel disease [64]. The genetic background seems to be particularly important in premature atherosclerosis development. In group of patients with the premature MI, the frequency of wild, heterozygous and mutant genotypes of C1562T and R279Q *MMP-9* polymorphisms were significantly different in comparison with the control group and the increased prevalence of heterozygous and mutant genotypes in patients with MI was noted [65]. In the Chinese population, the -1562CT and TT genotypes were more frequent than CC genotypes in patients with MI compared to the control group [66]. European patients with metabolic syndrome and the *MMP-9* -1572T allele are more susceptible to cardiovascular events. The observed vulnerability was enhanced additionally by increased total MMP-9 level. In patients without a metabolic syndrome, neither the T-allele occurrence nor MMP-9 activity were associated with the risk of an event [67]. The meta-analysis of literature concerning the association of *MMP-9* rs3918242 SNP and susceptibility to MI revealed that carriers of CT and TT genotypes were less susceptible to MI in comparison with CC homozygotes [68]. As regards *MMP-12* A/G polymorphism, carriers of G allele, which were characterized by lower MMP-12 promoter activity (-82AG and GG genotypes), had an increased risk of having MI and two or three vessel disease [56,69].

The imbalance between MMPs and TIMPs activity with an excessive fibrosis and depletion of correctly functioning cardiomyocytes in later post-MI stage results in maladaptive remodeling of myocardial tissue and leads to the development of progressive heart failure, which is another risk factor of thromboemolism in patients with AF [56,60,61]. The wide range of MMPs biological functions may contribute to the formation of optimal or insufficient scar in the process of post-MI healing. Disturbances in inflammatory response, appropriate fibroblasts proliferation and migration, angiogenesis, correct collagen synthesis and cross-linking, all of which are processes modulated by MMPs, may result in impaired scar formation, the weakening of the post-MI ventricle wall and finally in cardiac output decrease and heart failure symptoms. Additionally, as a result of cleavage of ECM compounds on smaller fragments by MMPs action, so-called matricryptins are released, which are involved in the proper wound healing and thus preserving LV structure [56,70,71]. Amongst MMPs involved in LV remodeling and HF development MMP-1, -2, -3, -7, -8, -9, -12,-14, -28 are mentioned [56].

Several SNPs in MMPs genes were reported as being associated with HF. It regards the *MMP-2* polymorphisms -1575 A/G, -1059 A/G, and -790 T/G implicating an increased risk of HF of any etiology and increased risk of cardiac mortality in the Brazilian population, as well those of African as Caucasian origin [72], similarly as -735 C/T polymorphism associated with HF in Czech patients [73]. Additionally, the -1171 6A variant of *MMP-3* associated with the lower protein activity than 5A variant was demonstrated as an independent predictor of HF-related deaths in patients with non-ischemic HF [74].

MMPs-enhanced expression and activity were demonstrated amongst pathophysiological mechanisms involved in hypertension development, progression and complications [75]. In animal models of hypertension, the association between elevated MMP-2 activity and altered conductance and resistance of arteries has been demonstrated. Additionally, biochemical changes in arterial walls resulted in higher susceptibility to vasoconstrictors and lower responsiveness to vasodilators [75,76,77]. In *Mmp-2* knockout mice treated with angiotensin-II, endothelial dysfunction was not observed, adversely to wild-type animals. These vascular alterations were reversible through the action of MMPs inhibitors or antioxidant drugs [78]. Suggested mechanisms of promoting arterial hypertension by MMP-2 excessive activity involve both the degradation of vasoconstrictors with the formation of more potent substances constricting the vessels and the cleavage of vasodilators [75]. Additionally, MMP-7 has been reported as contributing to vasoconstriction due to epithelial growth factor receptor (EGFR) activation [79]. In an animal rat model, the injection of MMP-7 and/or MMP-9 results in the rapid constriction of small vessels and arteries [80]. In human studies and clinical trials, in patients with arterial hypertension significantly higher serum levels of MMP-9 but a decrease in the results of hypertensive treatment were reported [81,82]. MMP-2 and MMP-3 are considered as enzymes initiating hypertension-related remodeling of arteries and large arteries walls stiffening. In hypertensive patients, an association between carotid-femoral pulse wave velocity (PWV) and plasma MMP-3 concentration was observed, whereas after 6 months of anti-hypertensive treatment, a significant decrease inMMP-2 and MMP-3 and elevation of TIMP-1 plasma levels, parallel to systolic BP lowering and PWV values were described [83]. As regards the genetic background of hypertension, *MMP-9*-1562C/T polymorphism was reported as an independent predictor of arterial hypertension on the basis of the observation, that in treatment-naïve, hypertensive T-allele carriers, the values of both systolic as diastolic BP, carotid-femoral PWV and serum MMP-9 concentrations were significantly higher than in CC-homozygotes [84]. The chronic target organs damage, which is the most unfavorable effects of hypertension, is mediated by several pathogenic mechanisms associated with the over-activation of pro-inflammatory molecules, growth factors, RAAS agents or reactive oxygen species, and frequently was observed in relation with increased MMPs plasma level (MMP-9, MMP-2, MMP-3 and MMP-1) [85]. Taking into consideration the meta-analysis conducted by Marchesi et al., MMP-2 and MMP-9 appear to be potential markers of cardiovascular remodeling, which underlies LV hypertrophy (LVH) and/or diastolic heart failure (DHF) in hypertensive patients [86]. Authors of another study demonstrated different plasma MMPs profiles in patients with hypertension complicated by LVH in comparison with patients suffering from DHF, apart from elevated levels of MMP-2 and MMP-9 in hypertensive patients with cardiovascular remodeling, with the significantly increased concentrations of MMP-3 and -7 and a decreased level of MMP-8 in the latter group of patients [87]. The role of MMP-2, MMP-9 in renovascular remodeling include both the protective effect depending on MMPs activation and degradation of excessively synthesized collagen in the early stages of HT-mediated renal damage and the adverse results leading to nephrosclerosis and chronic kidney disease in later stages [85,88].

Interestingly, also in obese people being in the group of extraordinary risk of early atherosclerosis development as well as recurrence of AF, the increased levels of MMP-9 were observed [89]. Similarly, MMP-9 serum levels were elevated in patients with confirmed atherosclerotic lesions in carotid arteries in comparison with healthy controls, whereas the level of MMP-1 and MMP-3 did not differ significantly between two evaluated groups [90]. In patients with type 2 diabetes (TDM2), which is a risk factor of atherosclerosis and elevates the risk of stroke in patients with AF, MMP-7 and MMP-12 plasma levels were increased in comparison with patients without glucose metabolism disturbances [91]. On the genetic level, the -1306 C/T *MMP-2* (rs243865), polymorphism located in the CCACC box influences the Sp1 binding site, which results in decreased activity of promoter in carriers of T allele and overexpression of MMP-2 protein in CC homozygotes, which was demonstrated as the factor contributing to increased susceptibility for TDM2 [92,93]. Another SNPs was demonstrated as being associated with the presence of TDM2 in humans are *MMP-2* -1575 G/A (rs243866) polymorphism and two *MMP-9* polymorphisms -1562 C/T (rs3918242) and +279 A/G (rs17576) [92,94,95].

In recent decades, several pharmacological agents were demonstrated as influencing the MMPs activity and/or regulating the MMPs/TIMPs balance. Amongst them are angiotensin-converting enzyme inhibitors (ACEIs), angiotensin receptor blockers (ARBs), statins, mineralocorticoid receptor antagonists (MRAs) antidiabetic and antiplatelets agents [96], all widely used in the treatment of diseases that are risk factors of stroke in patients with AF and included in so-called “upstream” therapy [97]. Current guidelines of management in AF indicate the role of preventing the occurrence of arrhythmia (primary prevention) as well as its reoccurrence and distant consequences (secondary prevention). In addition to strictly antiarrhythmic drugs, agents modifying atrial structural changes such as fibrosis, inflammation and oxidative stress and directly or indirectly affecting atrial ion channels are recommended [10,97]. MMPs inhibition seems to be potential target in upstream therapy. In a canine model, it was demonstrated that the suppression of MMPs with an experimental inhibitor limits atrial fibrosis and AF occurrence [98]. Additionally, in rats treated with doxycycline, which was described as the agent suppressing MMPs activity and down-regulating the transcription of MMPs mRNA [99], attenuation of the fibrosis-induced changes in atrial conduction and decreased susceptibility to AF were observed [100,101]. As MMPs activity and expression are increased by the angiotensin secretion [102], it should be expected that agents suppressing the action of angiotensin such as ACEI and ARB decrease MMPs activity, inhibit atrial fibrosis and remodeling, and so prevent AF occurrence. In vitro studies confirmed the favorable effect of molecules representing both groups on MMP-9 expression and activity [103]. It was observed that the treatment of hypertensive patients with candesartan or lisinopril results in the decrease inMMP-9 and the raise of TIMP-1 plasma levels [104]. The treatment of patients with CAD with enalapril and irbesartan exerts this same effect of reducing MMP-9 activity [105]. Moreover, telmisartan or enalapril administered in the acute phase of MI significantly decreased MMP-2 and MMP-9 activity [106], and treatment in patients with MI with valsartan or trandolapril for 12 months resulted in the suppression of MMP-9 plasma levels [107]. Despite the lack of randomized controlled trials (RCT) evaluating the impact of ACEI/ARB on new-onset AF, the results of retrospective analyses and trials, in which AF was pre-assumed as secondary endpoint, suggest the favorable effect in patients with co-existing LV dysfunction or hypertrophy as well as with subjects with hypertension [10,97].In the secondary prevention (in patients after ECV), encouraging results of small clinical studies were not confirmed in larger, prospective RCTs [10,97]

Additionally, aldosterone inducing fibrosis and inflammation plays a deleterious role in cardiac remodeling and AF development. On the contrary, aldosterone antagonists (spironolactone and eplerenone) exert the beneficial effect [108,109,110]. In an animal study, spironolactone ameliorated atrial structural remodeling in dogs with prolonged rapid pacing-induced AF. The treatment with spironolactone reversed the structural and functional changes of atria induced by 6weeks of rapid pacing, including the increased expression of MMP-9 and decreased level of TIMP-1. Similarly, the vulnerability to AF and arrhythmia duration time were reduced after spironolactone administration [111]. In patients suffering from CHF, therapy with spironolactone improved parameters of LV function in correlation with MMP-9/TIMP-1 ratio. Furthermore, MMP-1 and MMP-2 plasma levels were significantly decreased [112]. In subjects being at risk for HF development evaluated in prospective, spironolactone treatment could slow the progression to HF due to antifibrotic and anti-remodeling effects. One of biomarkers of fibrosis assessed in this study was MMP-2, the level of which significantly decreased from baseline to the last visit [113]. Additionally, the treatment with another MRA—eplerenonon—in an animal model of hypertensive HF resulted in the inhibition of MMP-2, MMP-12 and MMP-13 activities, as well as improvement of myocardial remodeling [114]. Sacubitril-valsartan, a combination newly introduced into the treatment of HF with reduced EF (HFrEF), significantly improves the left ventricle function and also leads to decrease inMMP-9 activity [115,116,117]. In patients with CHF (both with reduced and preserved EF) MRA, simultaneously with the improvement of other cardio-vascular outcomes, have reduced the new-onset AF occurrence [10,118,119]. Because AF is considered as a marker of HF severity, the beneficial effect of MRA on AF development may be caused indirectly by the heart function improvement, but at least partially may contribute to the inhibition of MMPs pro-fibrotic action. Further appealing candidates for upstream therapy are statins, inhibitors of hydroxyl-methylo-glutharylo coenzyme A (HMG-CoA) reductase, which are pleiotropic agents involved, amongst others, in inflammatory process. The key mechanism of the statin influence on MMP activity is the inhibition of nuclear-factor kappa B (NF-κB) which, apart from stimulation of various inflammatory cytokines expression, activates also MMP-9 expression [120]. In apoE-deficient rats typically developing ventricular hypertrophy and fibrosis, the treatment with simvastatin caused a reduction in MMP activity and prevented cardiac remodeling [121]. Furthermore, rosuvastatin inhibited *Mmp-2* and *Mmp-9* expression in LDL–receptor deprived animals [122]. In hypercholesterolemic patients, simvastatin, atorvastatin and rosuvastatin reduced MMP-9 activity and MMP-9/TIMP1 ratio [123,124,125]. However, in well-designed clinical trials, favorable effects of statins regarding reverse remodeling and AF prevention were not confirmed. Results of large registers suggest possible advantages of therapy with statins in particular groups of patients with AF already treated with beta-blockers [10,126].

As the relevance of drugs potentially inhibiting the atrial fibrosis in patients with AF is not definitely established, the wide field for further experimental and clinical investigations appears. Taking into consideration the role of MMPs in the process of fibrosis, the therapeutic approach should involve treatment with agents inhibiting MMPs.

### 3.3. MMPs in Stroke

As was presented above, the MMPs and TIMPs deregulation lays at the background of several risk factors more or less directly contributing to stroke. Moreover, when the stroke already occurs, MMPs exert an adverse impact in the acute phase and a beneficial influence in the post-stroke period [127]. In studies conducted in both animal and human brains, after stroke or after experimental middle cerebral artery occlusion (MCAO) in comparison with controls, increased levels of MMP-2, -3 and -9 were demonstrated [97,98]. MMP-9 is considered as the most involved in inflammatory response induced by cerebral ischemia. In the acute phase of stroke in rodent models, the increased levels of MMP-9 protein were detected within 2–24h after cerebral artery occlusion. Two days after stroke, MMP-9 and MMP-2 concentrations were significantly higher in ischemic lesions in comparison with non-ischemic lesions and, additionally, an elevated level of MMP-9 was found also in the ischemic penumbra [127,128,129,130,131,132]. This indicates a potential role not only in the acute phase of IS, but also in reperfusion injury and the progression of stroke. The observed coincidence with a disruption of the blood–brain barrier (BBB) and degradation of the neurovascular basal lamina, and the ECM depends on secretion of MMP-9 by endothelial cells and recruited leucocytes. The aberrant proteolysis results in parenchymal destruction of brain tissue, BBB opening and the rupture of brain vessels, which contribute, respectively, to cerebral infarct extension, cerebral edema and hemorrhagic transformation of stroke [127,133,134]. On the other hand, MMP-9 and MMP-2 inhibitors are able to antagonize the BBB damage in the experimental stroke model with fibrin-rich clot used to MCAO [135]. Additionally, levels of MMP-8 and MMP-8/TIMP-1 ratio assessed in a cross-sectional study in patients with first IS and in a control group became independently associated with stroke. The highest levels of MMP-2 amongst patients divided into subgroups according to subtype of IS were confirmed in patients with CES [136]. Contrary to the acute phase of stroke, in the recovery stage after IS, MMPs exert beneficial effects. In this phase, damaged compounds of ECM are digested and new tissues are formed. During the cleavage of ECM by mainly MMP-7 and MMP-9, but also MMP-1, -2, -3, -10, and -11, active forms of growth factors, which stimulate angiogenesis, vasculogenesis and neurogenesis are released. It contributes to proper tissue remodeling in the healing process. MMPs are also involved in the migration of neuronal precursor cells to areas damaged by stroke [127,131]. On the genetic level, several SNPs were identified as associated with *MMP* genes expression and/or MMPs activity and finally IS occurrence or outcome. The most intensively investigated were SNPs at *MMP-9* gene, which influenceMMP-9 activity and, respectively, cardiovascular mortality in patients with coronary heart disease [137], sex-related carotid atherosclerosis stiffness [138] and elevated risk of stroke [139]. Particularly, rs3918242 polymorphism (-1562C/T) present in the promoter region of *MMP-9* gene affects the structure of MMP-9 and enzyme activity. The result of polymorphism is the existence of promoter genotypes with low (C/C) or high (C/T, T/T) activity and an inhibited or enhanced expression of MMP-9 protein. When the T allele occurs instead the C allele, gene transcription is elevated, as well as enzyme synthesis and release, which provides excessive ECM degradation with adverse consequences such atherosclerotic plaque formation, rupture and reperfusion detriment [127,137]. In a Polish population of patients with IS (including CES), in comparison with a control group, a significantly higher frequency of T allele and genotypes CT + TT was shown. Authors noticed that patients with T allele were younger than patients with CC genotype at the moment of IS onset. Allele T and CT + TT genotypes appeared more frequently in patients with T2DM than in patients with normal range of plasma glucose [140]. In Chinese populations, originating from different regions, CC and CC + CT genotypes in comparison with TT genotype were related to significantly higher risk of IS. The CC + CT genotypes were associated with the increased IS risk in tobacco smokers and in patients with higher BMI [141,142]. In Indian patients with IS, CT genotype, TT genotype and T allele were significantly more frequent in patients with stroke, but *MMP-9* methylation was identified as a protective factor [143]. Taking into consideration the univocal research conclusions of studies evaluating the relationship of *MMP-9* SNPs variants with IS occurrence in various investigated populations depending on ethnics and even region, the meta-analysis including over 1600 cases and over 1500 controls was conducted. The analysis of seven selected studies revealed no significant correlations between MMP-9 -1562C/T polymorphism and the risk of stroke in any subgroups stratified according to ethnicity, age and type of stroke [144]. On the other hand, in a more recent meta-analysis, including almost 4000 Chinese patients with IS and group of age-matched controls, the correlation between the dominant model (alleles TT + TC versus CC) and IS occurrence was identified, but no significant differences in homozygous or heterozygous model was found [145]. Additionally, in a meta-analysis of 19 studies including Asians and White populations, a significant association of the rs3918242 polymorphism in overall population and the risk of IS was confirmed. The correlation remained significant in both males and females, patients aged ≥65 years and<65 years as well as in diabetic and the non-diabetic subgroups [146]. The rs17577 polymorphism, which exhibits genotypes AA, AG and GG and is in the complete linkage with rs3918242 polymorphism, can influence the structure of MMP-9 protein, although no significant correlation with the risk of IS was detected [147]. Otherwise, the SNP at loci rs3787268 *MMP-9* was correlated with increased risk IS. G/A and AA genotypes were associated with elevated and C/G genotype with reduced risk of IS occurrence [148]. Additionally, a synergistic effect of two-loci interactions between rs3918242 and rs3787268 promoting hemorrhage transformation in acute IS patients with atherothrombosis, small artery disease and CES was revealed [149]. Furthermore, in patients with AF,*MMP-9* gene polymorphism appears to be associated with the severity of stroke and the risk of early neurological deterioration (END). The occurrence of alleles rs3918242 CT/TT and rs3787268 AG/GG independently predicts the stroke severity, whereas alleles rs1056628 AC/CC and rs3918242 CT/TT were related to the elevated risk of END [150]. The rs1056628A/C variation affects the binding of micro-RNA (miR-491), which results in lower MMP-9 protein expression associated with the increased risk of IS in the case of rs1056628A allele occurrence in comparison with rs1056628C allele [151]. The higher susceptibility for stroke was also detected for polymorphic variants of another MMPs genes. Assessing SNPs in four genes (rs1799750 in *MMP-1*, rs243865, rs2285053, rs2241145 in *MMP-2*, rs17576 in *MMP-9*, rs660599, rs2276109, and rs652438 in *MMP-12*) and their interactions allows us to detect an association of rs17576 AG and GG genotypes with the increased risk of IS and significant interaction between *MMP-9* rs17576 and *MMP-12* rs660599 also correlating with the higher risk of IS [152]. It was observed that SNPs in -1607 1G/2G in *MMP-1* and -82 A/G in *MMP-12* genes were significantly associated with IS in whole evaluated population, with the -82 A/G polymorphism being a risk factor in European populations and *MMP-1* -1607 1G/2G and *MMP-12* -82 A/G in African populations [153]. An analysis of *MMP-1* (-1607 1G/2G) and *MMP-3* (-1171 5A/6A) polymorphisms in subtypes of IS revealed higher frequency of the 5A/6A+5A/5A genotypes and 5A allele in the overall investigated population with IS, whereas *MMP-1* 1G/2G and 2G/2G genotypes were associated with higher susceptibility for small-artery occlusion subtype and *MMP-3* 5A/6A+5A/5A genotypes enhanced the risk of large-artery atherosclerosis as well the first occurrence as recurrence, but not small-artery occlusion and cardioembolism [154,155]. The associations of MMPs genes polymorphisms with IS summarizes the meta-analysis including 29 studies. Authors indicated *MMP-9* (-1562C/T) and *MMP-12* (-1082 A/G) gene polymorphisms as being associated with an increased risk of IS, but the role of *MMP-1* (-1607 1G/2G), *MMP-2* (-1306C/T) and (-735C/T) as well as *MMP-3* (-1612 5A/6A) as the risk factors of IS was not confirmed [156]. Data regarding a clinical relevance of above-mentioned *MMPs* variants are presented in Table 1.

Taking into consideration the univocal and ambiguous results of studies investigating the potential influence of *MMPs* genetic polymorphisms on IS risk, authors of these studies suggest the role of large prospective studies and evaluation of SNP–SNP interactions as well as counteracting with environmental features.

Because of the significant relevance of MMPs activity in stroke development, course and outcome, the inhibition of MMPs with pharmacological agents seems to be a valuable therapeutic approach. Studies evaluating experimental MMPs inhibitors provided in animal models of stroke initially were encouraging. In rodents, early treatment with MMPs inhibitors [BB-94 (batimastat), GM6001] administered in the acute phase of stroke diminished edema and infarct size, reduced the number of hemorrhagic complications and promoted angiogenesis during recovery [127,139,157,158,159]. Prolonged inhibition of MMPs (days) or administration of agents such as BB-94, BB-1101 and FN-439 in the later period of stroke results in limitation of BBB leakage, but no benefits and even deterioration in neurologic and behavioral functions of tested animals were observed [160,161]. Carnosine, which inhibits MMPs activity by chelating zinc ions, was described as an agent reducing the ischemic injury and edema volume in a rat model of stroke. Carnosine injected intravenously assuaged brain damage in both permanent and transient ischemia in rats. The treatment was well tolerated and no toxic side effects were observed. The beneficial effect of carnosine is contributed to inhibition of tight-junction proteins damage, which results from the excessive activation of MMP-9 and MMP-2 in endothelial cells [162]. In animal studies, the effectiveness of doxycycline and minocycline, derivatives of tetracycline, in inhibiting MMP-2 and MMP-9 in a rat model of cerebral ischemia was confirmed [163,164]. In clinical trials, minocycline given alone or with tPA to patients with moderately severe acute IS appeared to be safe, well tolerated and even improved patients outcomes [165,166]. The role of minocycline in HT preventing is discussed in greater detail in the next section. Additionally, the role of statins in inhibiting of MMPs in IS was evaluated. Simvastatin administered during the first week of IS inhibited the serum MMP-9 activity, although the early treatment with simvastatin (started at first day after stroke onset) influenced neither MMP-9 nor TIMP-1 levels [167].

### 3.4. MMPs in Hemorrhagic Stroke Transformation

Of all consequences of cardioembolic event, hemorrhagic transformation (HT) of ischemic stroke seems to be the most severe form of cerebrovascular event. Importantly, the general cardiovascular risk is related with the increased occurrence of not only IS, but also HT [168]. As mentioned above, a synergistic effect of two-loci interactions between rs3918242 and rs3787268 promote the hemorrhage transformation of ischemic area in IS [149]. The relationship between MMP expression/polymorphism and risk of conversion of stroke area from ischemic to hemorrhagic one may have significant therapeutic implications. In stroke patients subjected to pharmacological recanalization with recombinant tissue plasminogen activator (rtPA), TT-genotype variant of both *MMP-9* -1562C/T polymorphism and of its inhibitory counterpart *TIMP-2* -303C/T was related with significantly increased risk of hemorrhagic stroke transformation [169]. In general, reperfusion-related secondary hemorrhage/hemorrhagic transformation of stroke area is a regularly described complication of pharmacological or mechanical recanalization [170,171]. Here, the disintegration of extracellular matrix [172], opening of blood–brain barrier [173,174] and dissolving of neurovascular unit [175] are the pathomechanisms commonly observed in hemorrhagic transformation of ischemic stroke (as reviewed in [157]) and increased activity of MMP, potentially released from neutrophiles by rtPA [176], for example, has been attributed to all these phenomena [177,178,179]. In both animal and human investigations of HT background, it was noticed that early HT (which occurred 18–24 h after IS onset) is related with MMP-9 released from leucocytes and brain—derived MMP-2. If the HT is considered as delayed (occurs after 18–24h of stroke onset) apart from MMP-2 and MMP-9, MMP-3 as well as endogenous tPA, are involved [180]. Of note, in clinical situations, not only a selective increase inMMP-9 activity but rather a disturbed balance of MMP-9/TIMP-1 ratio toward increased MMP activity was independently associated with relevant hemorrhagic transformation after rtPA treatment of IS [181]. In a retrospective analysis, increased MMP-9 was identified as one of independent predictors of adverse clinical course after mechanical major vessel recanalization in ischemic stroke, including futile recanalization [182]. In prospective cohort studies and meta-analysis, increased MMP-9 activity correlated well with both the risk of hemorrhagic transformation of IS [183,184,185] and with the early BBB disruption, as visualized by magnetic resonance imaging (MRI) sequences of fluid attenuated inversion recovery (FLAIR) [186]. Notably, enhancing of hemorrhagic stroke complication by anticoagulation and antiplatelet drugs seems to rely partially on MMP-9 activation or inhibition, since in experimental studies, implementing animal models of stroke and recanalization, administration of dabigatran or rivaroxaban and apixaban demonstrated its superiority over warfarin therapy as to the risk of secondary hemorrhage in relation with decreased MMP-9 activation after stroke and reperfusion [187,188,189,190].

As to the mechanic recanalization in IS, embolectomy/thrombectomy became an indispensable component of treatment strategies in cardioembolic stroke. The number of endovascular procedures grew exponentially, as did the number of hemorrhagic sequelae, in particular if recanalization is performed in a delayed manner [191]. In addition to the time window, kinetics of MMP-9 serum level was demonstrated as a fair predictor of hemorrhagic transformation in a population of large vessel occlusion treated by mechanical recanalization [192], although the isolated MMP-9 assessment on admission was of less prognostic value as to the HT risk in embolectomy group [193]. On the other hand, the successful surgical treatment of hemorrhagic stroke complication by means of surgical hematoma evacuation and decompressive craniectomy was related with decreased level of MMP-9 over the course of functional recovery [194].

Curiously, MMP-related pathomechanisms seem to be involved in vast group of distinct cerebrovascular abnormalities, not limited to hemorrhagic transformation of initially ischemic form of stroke. For instance, *MMP* polymorphism was associated also with an increased risk of rupture and hemorrhage in arteriovenous malformations (AVM) [195,196] but not in cerebral aneurysms [197,198]. Another potential pathomechanism, linking primary hemorrhagic stroke with MMP activity is the degradation of intravascular amyloid β deposits. In condition of cerebral amyloid angiopathy (CAA) or during aging, the accumulation of β-amyloid in the walls of cerebral vasculature weakens their mechanical resistance leading to microbleeds of full-blown form of intracerebral hemorrhage [179]. The MMP-related degradation of these deposits potentially precipitates fragility of vessels, thus promoting the occurrence of CAA-related ICH. Indeed, MMP-2 and -9 as well as membrane-type 1 MMP (MT1-MMP) are capable of degrading both in vitro and in vivo different forms of β-amyloid [199,200], including its vasculotropic mutants L22Q and E38V [201]. Importantly, both increased MMP activity and raised MMP/TIMP ratio were attributed to amyloidolysis in vitro [202], in animal studies [203,204,205] and associated with the occurrence of CAA-related hemorrhagic stroke in clinical conditions [206,207,208,209]. Based on the variety of the cerebrovascular conditions with potential MMP involvement, the general idea of dysregulation in MMP function as the factor increasing propensity for the incidence of these entities, or at least aggravating their course, may be delineated.

Importantly, the link between the hemorrhagic transformation of IS and MMP activity or MMP/TIMP-1 ratio makes MMP to an attractive target for experimental therapies. Indeed, several studies have described that deleterious HT effect, or that its risk may be limited by the targeting of MMP-9 activity by administration of minocycline [210,211,212] or by simvastatin or candensartan pretreatment [213,214]. Additionally, positive effects of mesenchymal stem cell treatment in experimental stroke and recanalization have been linked with reduced risk of HT due to reduced MMP-9 activation [215]. Accordingly, first clinical trials implementing MMP inhibitors (e.g., minocycline) in treatment of ICH or subarachnoid hemorrhage have been attempted [216,217].

## 4. Conclusions

The MMP activity disturbances and genetic variations lay at the background of pathological processes contributing to atrial fibrillation, which is the most important cause of cardioembolic stroke, and to hypertension, heart failure, vascular disease and diabetes representing the risk factors of cardioembolic stroke in patients with atrial fibrillation. MMPs influence the stroke course and complications. Hence, MMPs may be considered as the common link connecting the atrial fibrillation, risk factors of stroke and finally stroke occurrence.

Regarding the predictive role of the MMPs genetic polymorphism in the general population, large prospective studies are required, as well as the evaluation of SNP–SNP interactions and counteracting with environmental features. The role of MMPs as potential risk factors of cardioembolic stroke seems to be particularly valuable in patients clinically included into the group of low risk of cerebral event.

The issue of particular relevance that should be studied is the relationship of MMPs status and susceptibility to hemorrhagic transformation in the context of administered reperfusion and anticoagulant therapy.

## Figures and Tables

**Figure 1 ijms-24-03628-f001:**
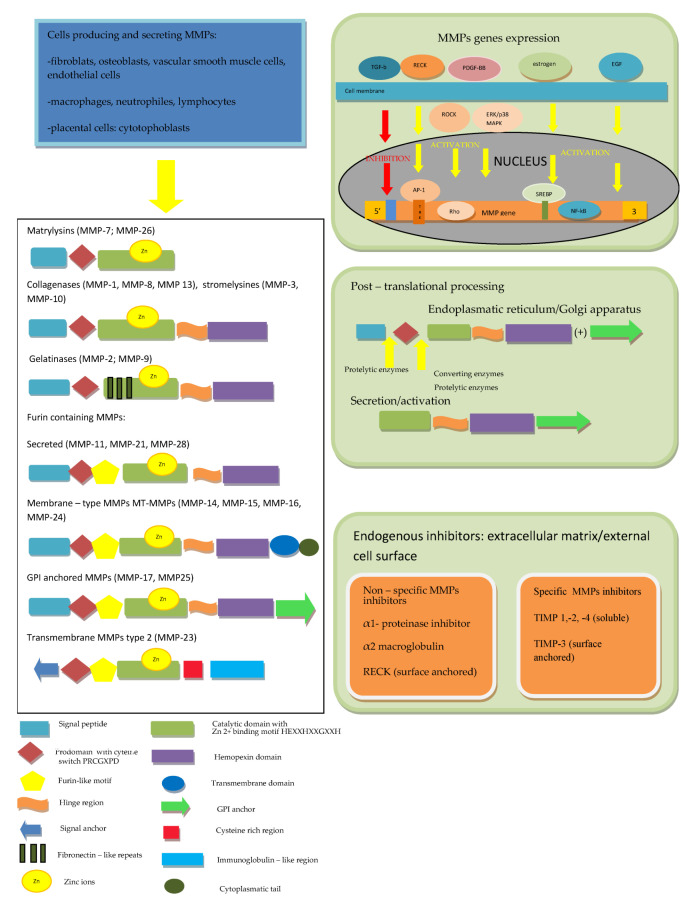
The MMPs’ structure and multifactorial and multi-level regulation of MMPs. According to [1,2,4,6,15,16,17,20,25].

**Table 1 ijms-24-03628-t001:** Clinical relevance of *MMPs* polymorphisms in stroke.

*MMP* Polymorphism	Relevance	Reference
rs3918242 (C/T) in *MMP-9*	Elevated risk of stroke	[139]
	T allele and CT + TT genotypes more frequent in IS (including CES), T allele more frequent in diabetic patients with IS	[140]
	CC and CC + CT vs. TT genotypes related to higher risk of IS	[141,142]
	CT, TT genotypes T allele more frequent in stroke	[143]
	TT + TC vs. CC related to IS occurrence	[145,156] (meta-analyses)
	CT/TT predicts stroke severity and elevate risk of END	[150]
rs3787268 (G/A) in *MMP-9*	GA and AA associated with elevated risk of IS	[148]
	AG and GG predict stroke severity	[150,152]
rs1056628 (A/C) in *MMP-9*	A allele vs. C allele increases risk of stroke	[151]
	CT/TT predicts stroke severity and elevate risk of END	[150]
-1607(1G/2G) in *MMP-1*	1G/2G genotype is risk factor of IS in African population	[153]
	1G/2G and 2G/2G genotypes associated with small artery occlusion	[154]
-1171 (5A/6A) in *MMP-3*	5A/6A and 5A/5A genotypes and 5A allele more frequent in IS	
	5A/6A and 5A/5A genotypes associated with large artery occlusion	[154,155]
-82 (A/G) in *MMP-12*	IS risk factor in European and African populations	[153]

## Data Availability

No new data were created or analyzed in this study. Data sharing is not applicable to this article.

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
