# Peer review of "Matrix Metalloproteinases in Cardioembolic Stroke: From Background to Complications"

_ijms, 2023, doi:10.3390/ijms24043628_

Round 1

Reviewer 1 Report

This is a review outlining the role of MMPs in cardioembolic stroke pathophysiology. There is a beginning portion discussing the MMP families. Then the authors transition to MMP association with LA remodeling which they believe is promoted by a mechanism of atrial fibrosis that they attribute to MMPs. The authors describe an association between MMP and cardioembolic strokes by describing lab experiments involving MMP knockout mice or population studies with particular MMP polymorphisms. The authors then move on to acute activation of MMPs in ischemic stroke in regards to BBB disruption and hemorrhagic transformation.

1. Numerous grammatical problems exist that hinder the impact of your interesting message. Here are examples:

              “MMP are a family of proteolytic enzymes of conservative structure and various functions in physiological and pathological conditions.”

              “MMPs are capable to cleave the peptide region….”

2. The authors extensive research into MMP polymorphisms is interesting and primarily promotes an idea of MMPs as chronic pathological activators of atrial fibrosis and possibly even metabolic syndrome risk factors. Would the authors venture to outline a more clinical-related proposal for dealing with this chronic feature of MMPs – i.e. atrial fibrosis and predisposition to AF? Would chronic suppression be justified or potentially harmful long-term?

3. The authors extensive background literature search is impressive. However, in the section 3.4, I would suggest keeping cerebrovascular pathologies separate to both hone the message and not introduce unnecessary information that might not be relevant for ischemic stroke. For example, if their manuscript is based on ischemic stroke, then I am not sure about the utility of providing information on AVMs, ICH/amyloid, and SAH – distinct pathologies whose mechanisms for introducing intracerebral hemorrhage is likely different from the hemorrhagic transformation mechanism of ischemic stroke.

4. Table 1 and 2 may be better served as one larger table.

Reviewer 2 Report

The review article entitled “Matrix metalloproteinases in cardioembolic stroke: from background to complications” is well presented on the importance of MMPs in mesothelioma.

Minor points

1.     In chapter 2, structure and regulation of MMPs are explained. It may be better if Table which summarizes the chapter 2 would be added.

2.     Line 48, IS account for → IS accounts for.

Line 68, patophysiology → pathophysiology.

Line 473, This indicate → This indicates.

Line 506, noticed a → noticed.

Line 512, patient → patients.

It is recommended to check English carefully.
